# Latent Representation Reorganization for Face Privacy Protection

## ABSTRACT

The issue of face privacy protection has aroused wide social concern along with the increasing applications of face images. The latest methods focus on achieving a good privacy-utility tradeoff so that the protected results can still be used to support the downstream computer vision tasks. However, they may suffer from limited flexibility in manipulating this tradeoff because the practical requirements may vary under different scenarios. In this paper, we present a two-stage latent representation reorganization (LReOrg) framework for face image privacy protection relying on our conditional bidirectional network which is optimized by using a distinct keyword-based swap training strategy with a multi-task loss. The privacy sensitive information are anonymized in the first stage and the destroyed useful information are recovered in the second stage according to user requirements. LReOrg is advantageous in: (a) enabling users to recurrently process fine-grained attributes; (b) providing flexible control over privacy-utility tradeoff by manipulating which attributes to anonymize or preserve using cross-modal keywords; and (c) eliminating the need of data annotations for network training. The experimental results on benchmark datasets have reported the superior ability of our approach for providing flexible protection on facial information.

## CCS CONCEPTS

• **Security and privacy** → **Privacy protections**; **Usability in security and privacy**.

## KEYWORDS

face image, privacy protection, recurrent, reorganization

## 1 INTRODUCTION

The issue of face privacy protection has aroused wide social concerns along with the increasing application of face images that carry lots of personal information (e.g. identity or religion) [31, 48, 56]. For example, the AI classifiers and DeepFake tools may easily read personal information and generate illegal clone avatars, which may bring about troubles (e.g. economic fraud) to individuals or organizations if the data are misused. This has led to the set up of more strict laws and regulations (e.g. GDPR, CCPA, PDPA and PIPA [1, 35, 38, 57, 58]) on data management. The immediate consequence is that people need to comply complicated legal or ethical constrains to avoid making troubles when accessing, using or disseminating the face data, which may block many important scientific researches

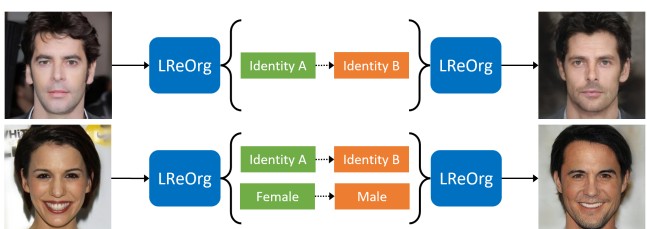

**Figure 1: Demonstration of LReOrg for face anonymization.**

or intelligent applications. One feasible way is to develop effective anonymization techniques to protect the sensitive attributes and preserve the desired non-sensitive ones because people may also expect that the protected data can still be useful (i.e. utility preservation) [11, 41, 45], where we collectively call all facial information as attributes (e.g. identity, gender and age). Such kind of techniques usually focuses on cheating both human and machine, and can be used to support various computer vision tasks (e.g. street-view map, autonomous drive and medical diagnostics [3, 30, 32, 60]) to clear up the restrains on privacy, ethics, laws and regulations.

Existing works show that privacy protection and utility preservation is a tradeoff problem, i.e. privacy-utility (PU) tradeoff, because both items are usually correlated [34, 38, 45, 60]. A higher performance of the former usually corresponds to a lower performance of the latter and vice versa. Recently, the generative methods [15, 23, 27, 46] receive increasing attention for producing realistic face images and achieving better PU tradeoff and various excellent methods have been proposed [5, 7, 16, 21, 22, 30, 38, 47, 54, 58] by replacing the original face image with a synthesized anonymous one, where some privacy protection strategies, like k-anonymity and differential privacy [12, 14, 40, 45, 49, 50, 54], were also studied to provide formal privacy guarantees. Although significant progresses have achieved, the up-to-date works still suffer from limited flexibility in manipulating the PU tradeoff under practical conditions and requirements by focusing on protecting the identity because some attributes may also become sensitive in some cases, especially when they are correlated to some special entities, events or activities, like religion, ethnic, laws and so on. Thus, it is reasonable and necessary to develop a more flexible mechnism for privacy protection.

To address the above problem, this paper presents a latent representation reorganization (LReOrg) framework for face privacy protection based on our conditional disentanglement-fusion network (CDFNet). LReOrg has several advantages for flexible anonymization: (1) it can enable users to recurrently process fine-grained attributes; (2) it can provide flexible control over PU tradeoff by manipulating which attributes to anonymize or preserve using cross-modal keywords; (3) it does not require any data annotations for network training. Existing works usually treat anonymization as a binary problem that hides the original identity and struggles to preserve the other attributes. Differently, we generalize this as a fine-grained problem by unifying privacy protection and utility preservation in a recurrent process by letting users to determine

how to perform anonymization (see Figure 1). To our best knowledge, this is the first time that the recurrent framework has been successfully used for deep face privacy protection. Our main contributions can be summarized as follows:

- A more resonable framework LReOrg is proposed for achiving a more flexible face anonymization by taking cross-modal keywords as fine-grained conditions.
- A CDFNet is designed to support forward feature disentanglement and backward feature fusion, which can recurrently support sensitive attributes anonymization and non-sensitive attributes recovery.
- We introduce a keyword-based swap training strategy supervised by using the CLIP model [44] and a multi-task loss.
- We rely on extensive experiments to quantitatively and qualitatively show the state-of-the-art performance of LReOrg by studying the privacy protection and utility preservation performances with respect to different attributes.

Note that CDFNet is built by following the architecture of Invertible Neural Network (INN) due to its excellent performance in image generation tasks [2, 25]. The work most related to ours is HiNet [25] which focuses on image steganography based on INN. CDFNet can be seen as a generalization of it, but they are quite different. First, CDFNet focuses on privacy protection, while HiNet focuses on image steganography. Second, CDFNet presents a new conditional version of INN based on cross-modal keywords. Third, the building blocks,their input and output are different. Besides, the network optimization method is also different.

## 2 RELATED WORKS

In this section, we discuss the most related works in contrastive language-image pretraining (CLIP) and face privacy protection.

**Contrastive Language-Image Pretraining.** Cross-modal vision and language representation has received lots of attention in various tasks these years, such as image caption and visual question answering. The success of Transformer [52] and BERT [9] has inspired many interesting works [43, 44, 55]. The recent CLIP model [44] has received wide attention by learning a multi-modal embedding space, which can be used to measure the sematic similarity between text and image. CLIP was trained on a 400 million sized dataset collected from the Internet, which has demonstrated powerful performances on various tasks and datasets. Due to the powerful ability of CLIP, we employ it as the cross-modal attribute discriminator to train our network to make it understand the relationships between text conditions and the anonymized face image.

**Face Privacy Protection.** Anonymization is regarded as an effective way to protect the privacy of face images, which is usually realized by de-identifying or hiding the original face identity while preserving the data usability. The commonly used simple anonymization methods, like blurring, pixelation and blacking out [24, 36], can destroy the data utility which receives increasing attention to enable the reusability of the anonymized data [20, 41, 45]. In recent years, the generative methods [15, 27, 28, 43, 46] exhibit promising performances on face image synthesis and anonymization by playing adversarial games [5, 21, 22, 30, 37–39, 47, 54, 59]. In [21], DeepPrivacy (DP1) relys on inpainting to generate anonymous face by blocking out the facial region. In [38], CIAGAN relys

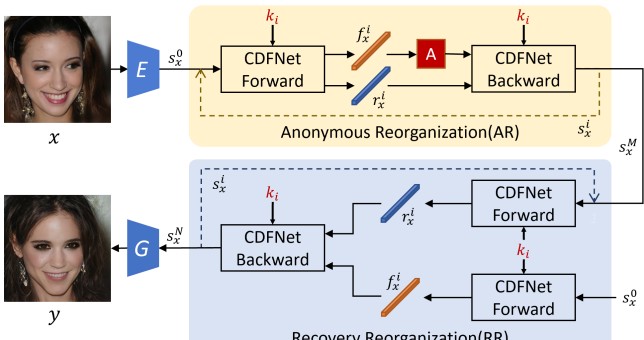

**Figure 2: The flowchart of our LReOrg framework.**

on masked image, landmarks and one-hot vector to perform conditional inpainting for face anonymization. In [54], IdentityDP relys on disentanglement and differential privacy [12, 49] for anoymous face synthesis after adding Laplace noise to identity feature. Although these methods can well protect the face identity, they suffer from some drawbacks on the naturalness of the anonymous face. In [22], DeepPrivacy2 (DP2) relys on continuous surface embedding and StyleGAN to further improve DP1. In [47], Clip2Protect relys on text-guided CLIP [44] and the StyleGAN latent space to generate anonymous face from the viewpoint of makeup, but it is time-consuming to finetune a new StyleGAN generator for each inference. In [30], LDFA presents a similar method as with DP1 by performing face inpainting based on latent diffusion model [46]. Although the image quality has greatly improved, these up-to-date works usually work in their predefined manner by mainly processing the identity, but lack of a mechanism to flexibly manipulate which attributes to anonymize or preserve in a more intuitive manner (i.e. poor flexibility). Differently, in this paper, we present a new privacy protection mechanism to perform fine-grained anonymization with cross-modal keyworks based on our bidirectional CDFNet, which can work recurrently according to practical requirements.

## 3 METHOD

In this section, we introduce the proposed LReOrg framework. We first have a brief overview in Section 3.1. Then, we introduce the proposed CDFNet network in Section 3.2. Finally, we present our traning strategy in Section 3.4.

### 3.1 Overview

In Figure 2, we plot the flowchart of LReOrg for face anonymization conditioned on the cross-modal keyword-based attribute representation in set $K = K_1 \cup K_2$, where $K_1 = \{k_i, 1 \le i < M\}$ is the sensitive keyword set, $K_2 = \{k_i, M \le i < N\}$ is the non-sensitive keyword set and each keyword corresponds to a different face attribute, such as 'identity', 'expression', 'gender' and 'age". The flowchart conissits of four steps. First, we generate a latent representation $s_x^0$ for a given face image $x$ by embedding it into some well defined latent space using encoder $E$. Second, we rely on the anonymous reorganization (AR) module to anonymize the sensitive attributes encoded in $s_x^0$ according to the keywords in $K_1$ (e.g. {'identity'}) by using our CDFNet and anonymizer $A$. Then, we rely on the recovery reorganization (RR) module to recover the non-sensitive attributes for the output $s_x^M$ of AR according to $K_2$ (e.g.

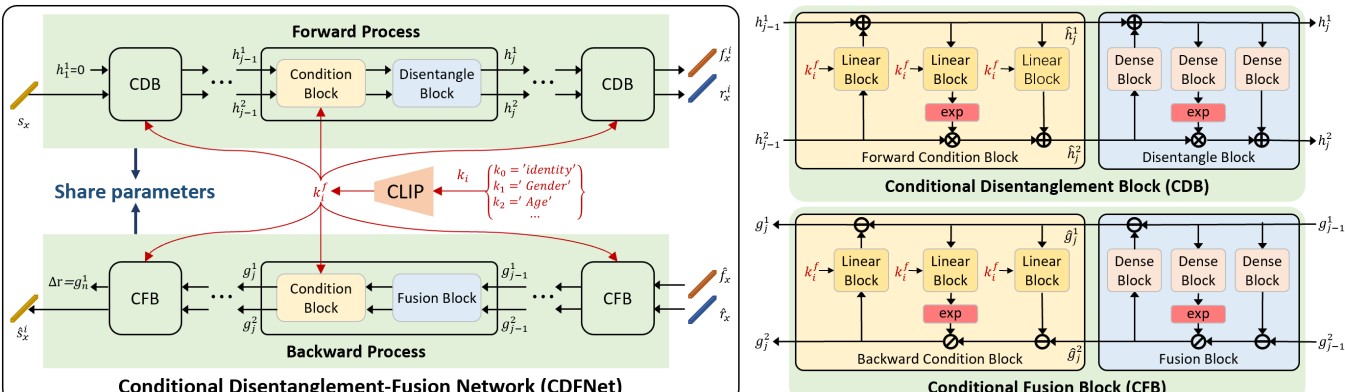

**Figure 3: The architecture of our CDFNet which consists of a forward process and a backward process. The forward process focuses on disentanglement and the backward process focuses on feature fusion and recovery.**

{'expression', 'gender' and 'age'}). Finally, we translate the output $s_x^N$ to a real face image $y$ by using generator $G$.

**Anonymous Reorganization.** This module focuses on sensitive attribute anonymization based on feature disentanglement. Given keyword $k_i \in K_1$, we first rely on the forward process of CDFNet to disentangle the latent representation $s_x^i$ into two features: the key feature $f_x^i$ and the residual feature $r_x^i$ of $s_x^i$. Second, we use our anonymizer A to process the sensitive information in $f_x^i$ and obtain $\hat{f}_x^i$. Then, $\hat{f}_x^i$ and $r_x^i$ are fused as a new latent representation $s_x^{i+1}$ in the backward process of CDFNet. Note that AR can not only separately anonymize single attribute but also jointly anonymize multiple attributes by recurrently processing them following the above steps, where $s_x^{i=0}$ is the initial input and $k_0$='identity' corresponds to the first attribute to be anonymized. The final output is the reorganized anonymous latent representation $s_x^M$.

**Recovery Reorganization.** This module focuses on non-sensitive attribute recovery in the latent space. Given keyword $k_i \in K_2$ ($M \leq i < N$), we first rely on the forward process of CDFNet to obtain the residual feature $r_x^i$ of $s_x^i$ and the key feature $f_x^i$ of $s_x^0$. Second, $r_x^i$ and $f_x^i$ are fused to a latent representation $s_x^{i+1}$ in the backward process of CDFNet. Similar to AR, RR can also jointly recover multiple attributes by recurrently processing them following the above steps, where $s_x^{i=M}$ is the initial input. The final output is the reorganized anonymous latent representation $s_x^N$.

## 3.2 The Proposed CDFNet

In this part, we introduce a new Conditional Disentanglement-Fusion Network (CDFNet) which is invertible and can receive bidirectional input. For easy understanding, we separate the network into bidirectional processes according to the data flow: the forward process and the backward process. The two processes share a same network structure as well as their building blocks and parameters, but differ only in the fundamental operations of arithmetic in their building blocks due to the opposite data flow. As shown in Figure 3, the forward process focuses on feature disentanglement by extracting a key feature $f_x^i$ and a residual feature $r_x^i$ from an input latent representation $s_x$ conditioned on the cross-modal keyword feature $k_i^f$ extracted by using the text encoder of CLIP. The backward process focuses on feature recovery by fusing a key feature $f_x^i$ and a

residual feature $r_x^i$ as a new latent representation $\hat{s}_x$ conditioned on $k_i^f$. To distinguish the two processes, we name the building block of the forward process as conditional disentanglement block (CDB), and the building block of the backward process as conditional fusion block (CFB). The corresponding linear and dense blocks in CDB and CFB share the same parameters, so the reversibility is entirely determined by changing operators.

**Conditional Disentanglement Block** can be understood as the building block of the forward process, which consists of a forward condition block (CB) and a disentangle block (DB). The DB block is borrowed from HiNet [25] for disentanglement. The forward condition block is developed in this paper to process the cross-modal conditions by following the INN rules [2, 10]. As shown in Figure 3, for the $j$-th CDB block in the forward process, the inputs are $h_{j-1}^1$ and $h_{j-1}^2$, and the outputs $\hat{h}_j^1$ and $\hat{h}_j^2$ are formulated as:

$$
\begin{aligned}
\hat{h}_j^1 &= h_{j-1}^1 + \phi_1(h_{j-1}^2, k_i^f), \\
\hat{h}_j^2 &= h_{j-1}^2 \otimes exp(\sigma(\phi_2(\hat{h}_j^1, k_i^f),, k_i^f)) + \phi_3(\hat{h}_j^1, k_i^f),
\end{aligned}
\tag{1}
$$

where $\phi_1$, $\phi_2$ and $\phi_3$ denote different linear blocks. Each one is realized with three fully connected (FC) layers: $\phi(a, b) = FC(concat(FC(a), FC(b)))$. The DB block is realized in a similar way as:

$$
\begin{aligned}
h_j^1 &= \hat{h}_j^1 + \psi_1(\hat{h}_j^2), \\
h_j^2 &= \hat{h}_j^2 \otimes exp(\sigma(\psi_2(h_j^1))) + \psi_3(h_j^1),
\end{aligned}
\tag{2}
$$

where $\psi_1$, $\psi_2$ and $\psi_3$ denote different densenet blocks [53].

**Conditional Fusion Block** consists of a backward conditional block and a fusion block, which can be obtained by reversing the data flow of CDB, where the backward conditional block (BCB) corresponds to the forward condition block (FCB) and the fusion block (FB) corresponds to the DB block of CDB. Given a pair of input $g_{j-1}^1$ and $g_{j-1}^2$, the output of FB is formulated as:

$$
\begin{aligned}
\hat{g}_j^2 &= (g_{j-1}^2 - \psi_3(g_{j-1}^1)) \otimes exp(-\sigma(\psi_2(g_{j-1}^1))), \\
\hat{g}_j^1 &= g_{j-1}^1 - \psi_1(\hat{g}_j^2).
\end{aligned}
\tag{3}
$$

Similarly, we formulate the output of BCB as:

$$
\begin{aligned}
g_j^2 &= (\hat{g}_j^2 - \phi_3(\hat{g}_j^1, k_i^f)) \otimes exp(-\sigma(\phi_2(g_j^1, k_i^f))), \\
g_j^1 &= g_j^1 - \phi_1(g_j^2, k_i^f).
\end{aligned}
\tag{4}
$$

After the inputs go through the sequence of CDB and CFB, the outputs of the last CFB block contains a information loss $\Delta r = g_n^1$ and a latent representation $\hat{s}_x^i = g_n^2$ which can be fed into generator G to generate a manipulated face image.

### 3.3 Anonymizer

We anonymize both the identity and the selected attribute information. Since they have different properties, we employ different strategies to protect them.

**Group-based Identity Anonymizer (GIA).** We impose a strict privacy protection strategy to ensure a better identity protection by using the well defined differential privacy theory [12, 49]. In the pre-processing step, the features in the latent space $S$ are clustered into $m$ groups $\mathcal{G} = \{\mathcal{G}_1, \mathcal{G}_2, \cdots, \mathcal{G}_m\}$, where we use $\bar{\mathcal{G}}_j$ to denote the average feature of the $j$-th group. Given an identity feature $f_x^i$ to be anonymized, we first utilize the classical exponential mechanism of differential privacy to sample a differentially private group $\mathcal{G}_u$ according to the distance between $f_x^i$ the $\bar{\mathcal{G}}_j$ under the constrained condition $a \le j \le b$ by considering the privacy and utility tradeoff. Then, in $\mathcal{G}_u$, we adopt the simple random sampling to choose one identity $\hat{f}_x^i$ to replace the original one. Because this process totally happens in the latent space, $\hat{f}_x^i$ can regarded as a virtual identity. Since differential privacy is resistant to any form of post-processing [12, 13, 49], the selection of $\hat{f}_x^i$ still follows differential privacy.

**k-farthest Attribute Anonymizer (KAA).** We adopt a simple method to anonymize each attribute. In the pre-processing step, we calculate the average feature as the representative of each sub-category of each attribute. Given an attribute feature $f_x^i$, we anonymize it by using the farthest average feature from the sub-categories following the k-anonymity rule [40, 50].

### 3.4 Keyword-based Swap Training

We train our CDFNet in a well defined latent space $S$ with the help of several pre-trained models, incuding generator G, latent space encoder MLP, image-text encoder CLIP $c(\cdot)$ and identity encoder $\rho(\cdot)$. As shown in Figure 4, we propose a keyword-based swap training strategy to optimize our network. First, we randomly sample a pair of latent representations $s_1$ and $s_2$ from $S$ in each training step. Then, we alternatively train CDFNet according to the cross-modal keywords $k_i$ sampled from $K$.

**Identity-oriented Swap (IOS) training.** IOS focuses on improving the identity disentanglement ability by performing identity swap training conditioned on keword $k_0 =$'identity'. First, we rely on the forward process of CDFNet to disentangle the key feature $f_1^k$ and the residual feature $f_1^r$ from $s_1$, which is the same for $f_2^k$ and $f_2^r$ from $s_2$. Second, we rely on the backward process of CDFNet to swap the key features of $s_1$ and $s_2$, resulting in two latent representations $s_1'$ and $s_2'$, where $\Delta r_1$ and $\Delta r_2$ are the information loss. By feeding $s_1$, $s_2$, $s_1'$ and $s_2'$ to generator G separately, we obtain four face images $I_1$, $I_2$, $I_1'$ and $I_2'$. Our network is optimized by using:

$$L_{IOS} = \lambda_1 L_I + \lambda_2 L_p + \lambda_3 L_f + \lambda_4 L_r. \tag{5}$$

where $L_p = \mathbb{E}[\eta(I_1, I_1') + \eta(I_2, I_2')]$ is the VGGFace based perceptual loss [6, 26, 42] and $L_r = \mathbb{E}[\|r_1\|_1 + \|r_2\|_1]$ is the information loss. $L_I$ is identity feature loss

$$L_I = \mathbb{E}[d(I_2, I_1') + d(I_1, I_2') - d(I_1, I_1') - d(I_2, I_2')], \tag{6}$$

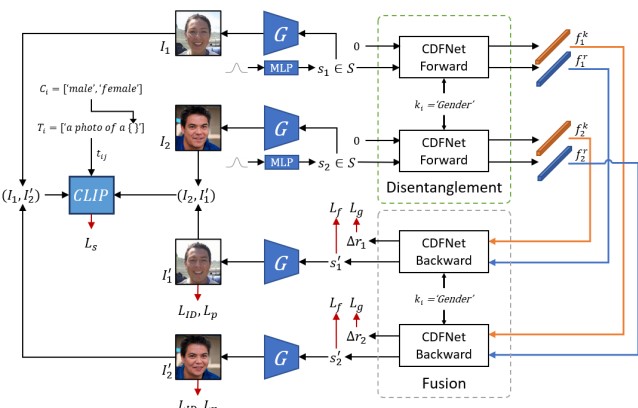

**Figure 4: The keyword-based swap training strategy.**

where $d(x, y)$ is the cosine feature distance between $x$ and $y$. The latent representation loss $L_f$ is defined as

$$L_f = \mathbb{E}[\|s_1 - s_1'\|_2 + \|s_2 - s_2'\|_2]. \tag{7}$$

**Attribute-oriented swap (AOS) training.** AOS focuses on improving the attribute disentanglement ability by performing attribute swap training conditioned on keword $k_i$, $i > 0$. The training process is the same as that of IOS, but the loss function is different

$$L_{AOS} = \alpha_1 L_{I'} + \alpha_2 L_s + \alpha_3 L_f + \alpha_4 L_r, \tag{8}$$

where $L_{I'} = \mathbb{E}[d(I_1, I_1') + d(I_2, I_2')]$ is the identity preservation loss. To associate the correspondence between facial attribute with the keyword $k_i$, we formulate the attribute swapping loss $L_s$ as an adversarial game

$$L_s = \mathbb{E}\Big[ -\sum_{j=1}^{|C_i|} H(I_1, I_2, I_1', I_2', t_{ij})\Big], \tag{9}$$

by taking the cross-modal CLIP as discriminator, where

$$H(I_1, I_2, I_1', I_2', t_{ij}) = D(I_1, t_{ij})\log(D(I_2', t_{ij})) + \\ D(I_2, t_{ij})\log(D(I_1', t_{ij})). \tag{10}$$

As shown in Figure 4, $t_{ij}$ is a sentence by filling $c_j \in C_i$ to $T_i$ (e.g. $t_{ij} =$ 'a photo of a male.'), $C_i$ is the fine-grained attribute set of the $k_i$ (e.g. $C_i =$ 'male', 'female') and $T_i$ is the $i$-th sentence template (e.g. $T_i =$ 'a photo of a {}.').

## 4 EXPERIMENTS

In this section, we perform quantitative and qualitative experiments to show the effectiveness of the proposed approach. More details and results are presented in the supplementary material.

### 4.1 Implementation Details

**Settings.** We use the pre-trained StyleGAN2 [27, 28] and GAN Inversion [51] to build the latent space because it would favor attribute disentanglement, where the latent codes $s_1$ and $s_2$ in Figure 4 are generated by the MLP layer of StyleGAN. StyleGAN is used as G and GAN inversion is used as the encoder E in Figure 2. Note that the face images we used in the training process were sampled or synthesized from the latent space of StyleGAN, which would favor anonymization. Table 1 presents some representative attribute

**Table 1: Example of the keyword set.**

| Keyword $k_i$ | Content Set $C_i$ |
|---|---|
| 'Gender' | {'male','female'} |
| 'Age' | {'young', 'old'} |
| 'Expression' | {'smiling', 'no smiling'} |
| 'Makeup' | {'heavy', 'no'} |
| 'Color' | {'blond', 'black', 'brown', 'gray'} |
| 'Curly' | {'curly', 'straight'} |
| 'Length' | {'long', 'short'} |

based keywords. For easy and fair comparison, we employ the inner face attributes {'identity', 'Gender', 'Age', 'Expression', 'Makeup'} for anonymization and recovery by setting $K_1$ ={'identity'} and $K_2$ ={'Gender', 'Age', 'Expression', 'Makeup'} by default, but they can be changed under different scenarios. Since the attributes 'Color', 'Curly' and 'Length' are related to the outer regions of the face, we advocate to let the users determine how to use them. We train our network on 256×256 images by using Adam optimizer ($\beta_1$ = 0.5 and $\beta_2$ = 0.999) with learning rate of $1e^{-5}$. We set $\lambda_1$ = 24, $\lambda_2$ = 1.2, $\lambda_3$ = 3.0, $\alpha_1$ = 0.2, $\alpha_2$ = 30, $\alpha_3$ = 3.0 and $\alpha_4 = \lambda_4$ = 10.

**Datasets.** The CelebA-HQ [33] and LFW [19] datasets are employed for evaluation. **CelebA-HQ** contains 30,000 face images from 6,216 identities, where 5,000 images are employed as the test set and the remaining are used for training. **LFW** consists of 13,233 face images from 5,749 individuals, where 5,000 images are used for test. We train our model on CelebA-HQ and evaluate it on all.

**Baseline Methods.** We compare our approach with the following representative and up-to-date methods, including: the classical generative methods CIAGAN and DeepPrivacy (DP1) [21]; the latest blurry method DarBlur [24] and DeepPrivacy2 (DP2) [22]; the latent represent methods LDFA [30] and Clip2Protect [47]; and the differentially private identity disentanglement method IdentityDP [54]. Since IdentityDP also employed differential privacy for anonymization in the feature space, we adjust our approach using the same manner as a baseline (denoted as IdentityDP) to show the effectiveness of our group-based identity anonymizer.

**Evaluation Criteria.** We evaluate our approach for privacy protection and utility preservation. For privacy protection, we evaluate the protection success rate (PSR, the higher the better) which is calculated as the percentage of protected faces missclassified by the face recognition tools, where the pre-trained ArcFace [8] and AdaFace [29] models are used. Face alignment [4] is used to detect face and calculate the detection rate (the higher the better). We use Fréchet Inception Distance (FID) [18] to evaluate the image quality (the lower the better). We evaluate the attribute preservation rate (APR) by using pre-trained classifiers, the higher the better.

## 4.2 Main Results

In this part, we show the performance of our approach by comparating with existing methods from different viewpoints.

**Protection and Preservation.** We hope that the anonymized face images can still be detected with a high protection successful rate, which means that a good anonymization method should have high face detection rate and low PSR rate. We first compare our results with the representative basic methods DeepPrivacy [21] and CIAGAN [38] in Table 2. The detection rates of all the methods reach 100%. The PSR rates of our results are higher than DeepPrivacy and

**Table 2: Comparison with the representative DeepPrivacy and CIAGAN methods on CelebA-HQ.**

| Method | PSR (%) ↑ | | Detection (%) ↑ | APR (%) ↑ | FID↓ |
|---|---|---|---|---|---|
| | Arcface | Adaface | | | |
| Original | 0 | 0 | 100 | 92.2 | 5.65 |
| CIAGAN | 97.4 | 97.3 | **100** | 74.0 | 102.8 |
| DP1 | 95.1 | 95.9 | **100** | 73.9 | 53.3 |
| Ours | **98.4** | **98.8** | **100** | **82.0** | **40.1** |

**Table 3: Comparison with the blurry methods.**

| Method | PSR ↑ | | Detection ↑ | APR ↑ | FID↓ |
|---|---|---|---|---|---|
| | Arcface | Adaface | | | |
| Blurring | 97.0 | 98.1 | 93.2 | 66.3 | 57.7 |
| DartBlur | **99.9** | **100** | 97.8 | 64.2 | 128.2 |
| Ours | 98.4 | 98.8 | **100** | **82.0** | **40.1** |

**Table 4: Comparison with the disentanglement method.**

| Method | PSR ↑ | | Detection ↑ | APR ↑ | FID↓ |
|---|---|---|---|---|---|
| | Arcface | Adaface | | | |
| IdentityDP | 87.1 | 87.9 | **100** | 81.5 | 53.9 |
| Ours | **98.4** | **98.8** | **100** | **82.0** | **40.1** |

**Table 5: Comparison with the SOTA methods.**

| Method | PSR ↑ | | Detection ↑ | APR ↑ | FID↓ |
|---|---|---|---|---|---|
| | Arcface | Adaface | | | |
| DP2 | 96.9 | 96.8 | 99.9 | 77.0 | 16.0 |
| LDFA | 87.6 | 88.7 | 98.5 | 83.6 | **8.1** |
| CLIP2Protect | 44.3 | 42.2 | **100** | **86.9** | 41.0 |
| Ours | **98.4** | **98.8** | **100** | 82.0 | 40.1 |

CIAGAN by using both Arcface and FaceNet. The average APR and FID scores of our result also outperform DeepPrivacy and CIAGAN, which stay close to the baseline results of the original data.

**Comparison with blurry methods.** In Table 3, we compare our method with the tradtional Blurring method and the latest generative DartBlur method [24]. It is obvious that both Blurring and DartBlur have very high PSR rates for Arcface and Adaface, but their attribute preservation rates and FID scores are not ideal. In contrast, our resutls show competitive PSR performances with much better APR and FID scores.

**Comparison with disentanglement method.** In Table 4, we compare our method with the representative feature disentanglement method IdentityDP which protect face identity by adding Laplace noise following differential privacy. The results show that simply adding Laplace noise may not work effectively on identity protection and may also affect the data utility on APR and FID.

**Comparison with the latest methods.** In Table 5, we compare our method with the closely related state-of-the-art (SOTA) methods. DP2 [22] works with StyleGAN2 generator, which is the improved version of DP1 [21]. LDFA [30] can be also be seen as the improvment of DP1 by using latent diffusion model. CLIP2Protect [47] is built based on CLIP and StyleGAN2. Compared with DP1 in Table 2, DP2 has significant performance improvement on both

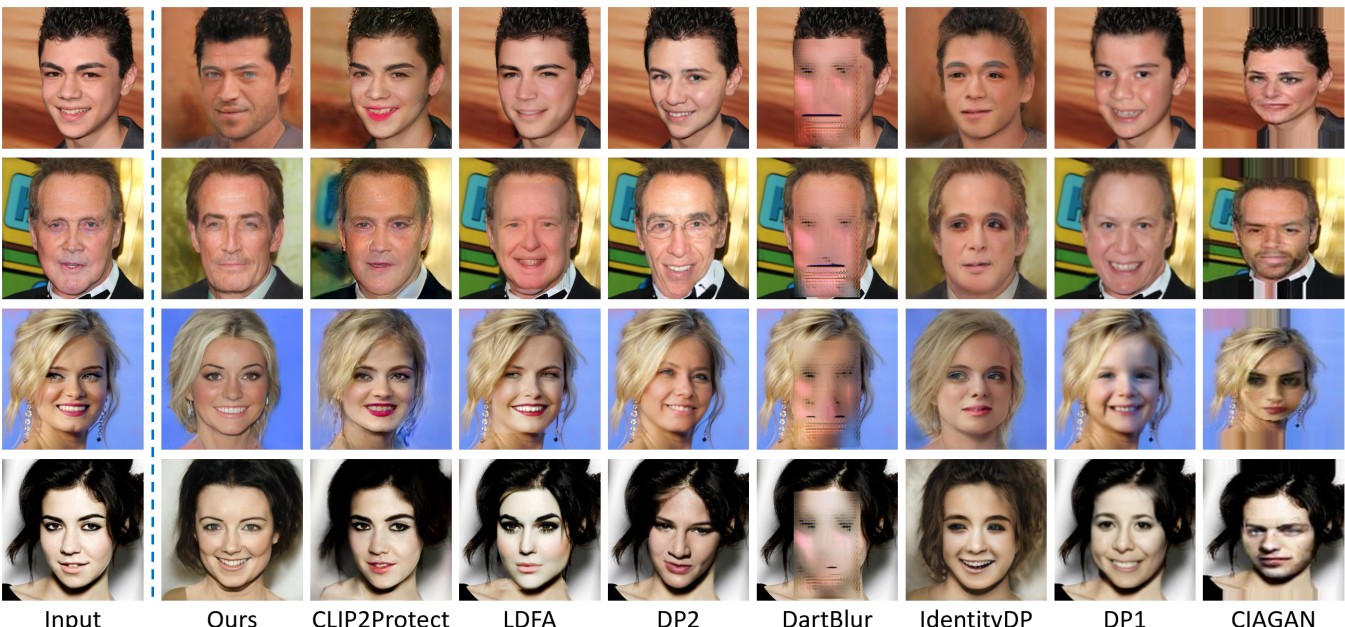

| Input | Ours | CLIP2Protect | LDFA | DP2 | DartBlur | IdentityDP | DP1 | CIAGAN |

Figure 5: The visual comparison of our LROrg results with that of SOTA. The first column presents the original input faces.

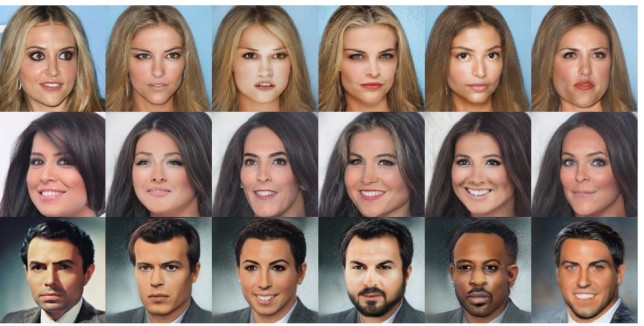

Figure 6: Example of the generated diverse results for the faces shown in the first colum.

privacy protection and utility preservation. The excellent FID performance of LDFA can be contributed to the denoise ability of the diffusion models. Both DP2 and LDFA suffer from some performance drop on face detection. The excellent attribute preservation performance of CLIP2Protect can be contributed to the finetuned generator for each input image, but the computational costs for inference is significant high and its identity protection ability is limited. In contrast, our method show the best identity protection performances with limited performance decrease on APR and FID.

**Image Quality and Diversity.** In Figure 5, we present some representative visual results. DartBlur may easily damage the key contents of face images, leading to significant degraded image quality. LDFA and CIAGAN may suffer from some rectangle effect. IdentityDP may suffer some distortions. The results of CLIP2Protect have a high probability to look similar as the input face except regardless of makeup, which cannot visually protect the face. DP1, DP2 and Ours show comparable good visual image quality and they show significant differences with the original input data. Since our method has no additonal operations on background in the latent

space, it cannot well preserve the original background, but it can be recovered by employing anothter recovery step following [38]. In Figure 6, we show that LROrg can also generate diverse anonymization results by default, which can be contributed to the random mechanism used in our group-based identity anonymizer. One can observe that the generated faces are anonymizd and look different from each other.

Table 6: The evaluation results on LFW dataset.

| Method | PSR ↑ | | Detection ↑ | APR ↑ | FID↓ |
|---|---|---|---|---|---|
| | Arcface | Adaface | | | |
| CIAGAN | 96.9 | 97.4 | **100** | 77.0 | 29.5 |
| DP1 | 92.4 | 94.7 | **100** | 83.0 | 53.7 |
| DartBlur | 98.0 | 99.0 | 99.2 | 78.6 | 59.3 |
| DP2 | 94.6 | 96.4 | **100** | 83.8 | 52.2 |
| LDFA | 93.4 | 94.9 | 99.2 | **85.0** | **5.2** |
| Ours | **99.6** | **99.6** | **100** | 78.8 | 73.9 |

**Transfer Capability.** We show the transfer capability of our method on the LFW dataset by using the pre-trained model on CelebA-HQ. We report the results in Table 6. It is obvious that Our method show consistent results as with that in previous experiments, which again reflects the superior performance of our method. The latent information loss of the StyleGAN latent space would decrease the performance of our FID score, which is a limitation of our method. Since all the evaluation can achieve almost 100% face detection rate, we no longer report them next.

**Personalized Protection.** We regard identity as one attribute so that users can flexibly choose which attribute to preserve or not according to the practical applications. In Figure 7, we present several examples of our personalized fine-grained anonymization process by removing or preserving some attribute according to the keyword conditions. It is obvious that our method can flexibly

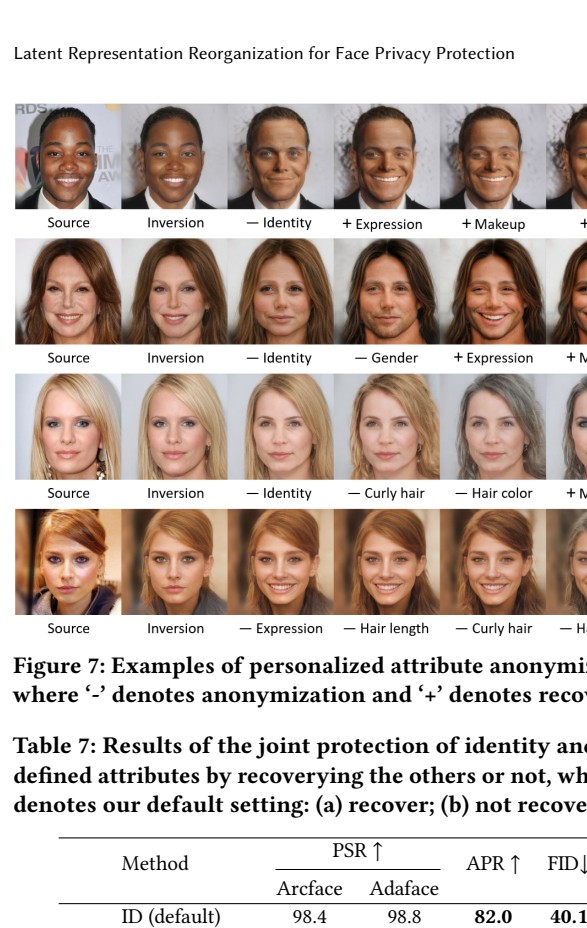

Figure 7: Examples of personalized attribute anonymization, where '-' denotes anonymization and '+' denotes recovery.

Table 7: Results of the joint protection of identity and user-defined attributes by recoverying the others or not, where ID denotes our default setting: (a) recover; (b) not recover.

| | Method | PSR ↑ | | APR ↑ | FID↓ |
|---|---|---|---|---|---|
| | | Arcface | Adaface | | |
| | ID (default) | 98.4 | 98.8 | **82.0** | **40.1** |
| (a) | ID+Gender | **99.4** | **99.7** | 72.4 | 60.5 |
| | ID+Age | 98.5 | 99.2 | 68.8 | 62.3 |
| | ID+Expression | 99.0 | 99.1 | 53.4 | **52.0** |
| | ID+Makeup | 98.6 | 98.8 | 79.6 | 53.8 |
| (b) | ID+Gender | 99.6 | 99.7 | 39.9 | 53.9 |
| | ID+Age | 99.4 | 99.5 | 47.7 | 47.5 |
| | ID+Expression | 99.3 | 99.5 | 47.3 | **35.9** |
| | ID+Makeup | 99.0 | 99.0 | 51.4 | 40.3 |

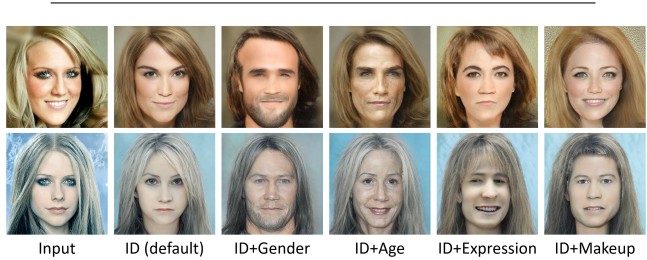

Figure 8: Visual results of the joint protection of identity and user-defined attributes while preserving the others.

anonymize and recover the given attributes according to the keyword conditions, resulting in realistic and desired face images. In Table 7, we report the quantitative evaluation results of protecting the identity and one user-defined attribute, such as ID+Gender. One can observe that, compared with our default settings, the joint protection strategy can further improve the PSR performance, but the data utility may suffer from different extent of drops. Without recovering the attributes, (b) suffer from lower APR and FID scores. Figure 8 demonstrates the corresponding visual examples.

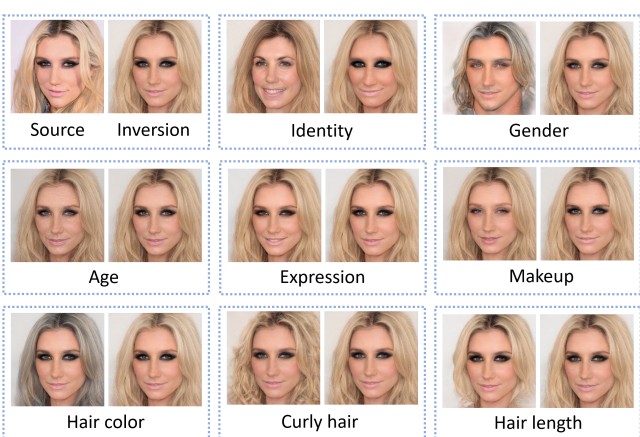

Figure 9: Illustration of the reversible abiilty of our approach after anonymization. In each face pair, the left one denotes anonymized version and the right one denotes the recovered version. Inversion means the face is reconstructed from GAN Inversion using StyleGAN.

Table 8: The recovery rate of each attribute.

| Attribute | identity | Gender | Age | Expression | Makeup |
|---|---|---|---|---|---|
| Rec rate | 59.5 | 99.0 | 84.7 | 81.4 | 84.1 |

Table 9: Ablation studies on the anonymization mechanisms.

| Method | PSR ↑ | | APR ↑ | FID↓ |
|---|---|---|---|---|
| | Arcface | Adaface | | |
| k-anonymity | 98.2 | **99.1** | 64.3 | 69.0 |
| Ours | **98.4** | 98.8 | **82.0** | **40.1** |

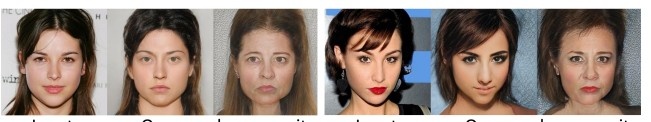

Figure 10: Viusal comparison with k-anonimity.

**Reversibility.** Since LROrg is built based on cINN [2], the protection is theoretically reversible. In Figure 9, we demonstrate the anonymization-recovery process for several representative attributes. The results show that our approach can well recover the anonymized attributes. In Table. 8, we report the recovery rate of different facial attributes after anonymization. The identity recovery ability suffer from significant drops. The reason may lie in the information loss during the disentangle-fusion process of our CDFNet as well as the information loss between StyleGAN and GAN Inversion, which is a limitation of our method to be addressed in the follow up works.

## 4.3 Ablation Studies

In this part, we conduct ablation studies to show the ability of our framework by varying the configurations.

**Anonymization Strategy.** For identity anonymization, we compare our differential privacy based method with that of the k-anonimity strategy [50] by using the farthest group center for identity anonymization to ensure a good protection. The results in Table

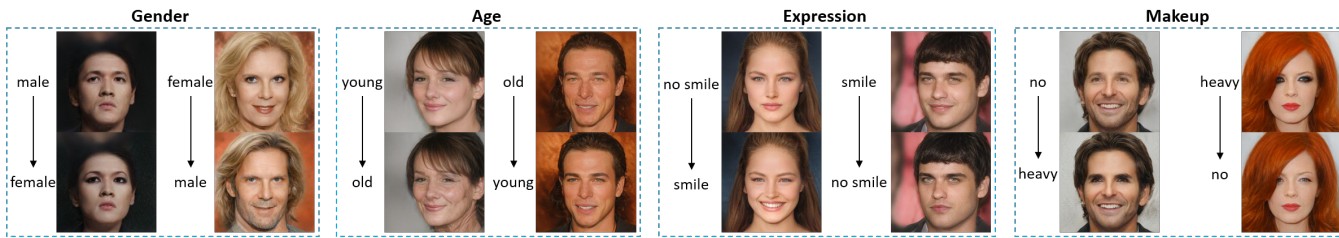

**Figure 11: Demonstration of non-identity attribute anonymization.**

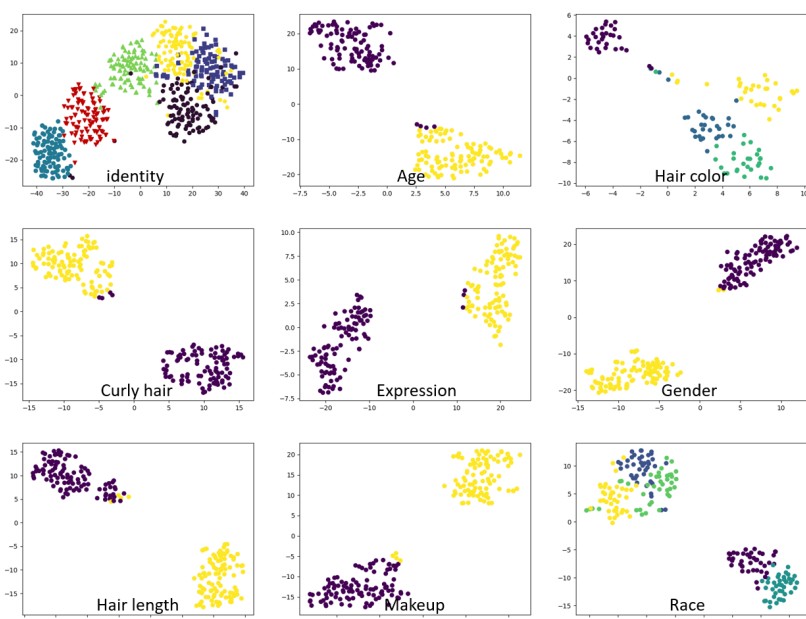

**Figure 12: Demonstration of the clustering performances of the disentangled features in the latent space.**

**Table 10: Impact of non-identity anonymization on PSR.**

|         | Gender ↑ | Age ↑ | Expression ↑ | Makeup ↑ |
|---------|----------|-------|--------------|----------|
| Arcface | **69.2** | 27.8  | 27.9         | 22.0     |
| Adaface | **51.0** | 13.6  | 11.5         | 9.5      |

9 show that our method not only has significant advantage over the k-anonymity strategy on APR but also outperforms k-anonymity on PSR. Figure 10 visually illustrates that the k-anonymity may suffer from more utility drops than ours, like age.

**Impact of Non-Identity Anonymization.** As shown in Figure 11, we also wonder the impact of non-identity attribute anonymization by processing one attribute each time without recovery. Table 10 reports the result. One can observe that anonymizing gender can produce much higher PSR scores than processing age, expression and makeup, which indicates that gender share more correlations with the face identity, which would makes it harder to preserve the gender attribute in privacy protection. And changing the makeup attribute may have the least impacts on identity protection.

**Latent Feature Distributions.** We studied the features disentangled by our CDFNet by clustering them into different groups. According to the plots shown in the Figure 12, we can observe that our disentangled features show good clustering performances for different kinds of attributes. This reveals that our model has good representation ability.

## 5  CONCLUSION

These years, the issue of face privacy protection has received increasingly attentions. In this paper, we present a keyword conditioned LROrg framework for fine-grained face privacy protection. On top of extensive experiments, we have verified the state-of-the-art performances of LROrg on achieving a better privacy-utility tradeoff, where the protection ability can achive further improvement by flexibly manipulating which attribute to anonymize or preserve according to practical requirements. We also find that gender is closely correlated with face identity which may inspire follow up works in aonymization. In comparison with previous methods, our solution is more flexible and effective by working in a distinct recurrent manner.

Although LROrg does not rely on data annotations, it suffers from the problem of incomplete latent space which is built by using StyleGAN and GAN Inversion. This would lead to the problem of information loss and further affect image synthesis. Besides, our CDFNet model may also suffer from the same problem, which would limit its performance. We will explore to address the problems in the future work.

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
