# OpenReview forum: "Latent Representation Reorganization for Face Privacy Protection"
_acmmm.org/ACMMM/2024/Conference — MM2024 Poster_

### Official Review · Reviewer_T6kt · 2024-05-04

**Rating:** 3
**Confidence:** 3

**Summary:**

This paper presents a latent reorganization framework for flexible face privacy protection, which can control which attributes to anonymize or preserve. A CDFNet is designed to  recurrently support sensitive attributes anonymization and non-sensitive attributes recovery.   A keyword-based swap training strategy is introduced by using the CLIP model and a multi-task loss.  The experimental results on benchmark datasets
 reported the superior ability of the method to provide flexible protection.

**Strengths:**

The research problem is meaningful and important.

Technological innovation is sufficient, which contains  a new network and a new training strategy.

The experiment is rich, which incorporates video presentations and feature visualisations

**Limitations:**

1. **De-identification does not work perfectly in this paper.** De-identification is the most demanded function in real-world applications, but the method in this paper does not allow for better performance in attribute preservation. As shown in Fig. 1 and Fig. 5, **other attributes of the face image are changed** when the identity is modified, such as background, hairstyle, and so on. However, existing advanced de-identification methods [1-2] have been able to preserve these attributes and only modify the identity region.  **Although this method supports the attribute recovery process, it is laborious for the user.** The user need to observe the protection results and explicitly indicate the retention of multiple attributes.  More unfortunately, the face images has more than 40 properties that are hard to list.

      [1] 2021ICCV-Personalized and Invertible Face De-Identification by Disentangled Identity Information Manipulation

      [2] 2020ECCV-Password-conditioned Anonymization and Deanonymization with Face Identity Transformers


2. **Why the keywords were used was not clearly explained.** What are the advantages of using text keywords over binary labels.  On my understanding, the keywords just go to guide the network to decouple the corresponding features, but simple numeric labels can go to do that as well. For example, using 10000 for identity change and 0001 for gender change. Furthermore, the paper does not compare with existing methods that use the label de-identification [3].

      [3] 2021TCSVT-The UU-Net_ Reversible Face De-Identification for Visual Surveillance Video Footage

3. **The effectiveness of flexible protection is doubtful.**

    **(1) Failure to achieve identity-preserving attribute protection.**  Identity-preserving attribute protection is also a mainstream class of face privacy protection methods. According to the description of this paper, the proposed method should be applicable to such methods as well, i.e., only anonymize the attributes and recover the identity.  However, this paper does not validate it, and also according to the description in Table 8, the method in this paper does not seem to be able to recover identity.

     [4]2020TIP-PrivacyNet: semi-adversarial networks for multi-attribute face privacy

     [5]2023TIFS-RAPP: Reversible privacy preservation for various face attributes;

   **(2) Restoring other attributes may affect the already anonymized attribute.** Correlations between attributes are not considered in this paper. In fact, when the gender attribute is restored, there is a high probability that it will affect previously anonymised attributes, which include 'Makeup', 'Color', 'Curly'. This is because these attributes are extremely correlated with gender.

4. **Unreasonable selection of baseline methods and inadequate development of flexible protection experiments.**  （1）The goal of CLIP2Protect is to generate adversarial faces (with insignificant changes), whereas this paper directly modifies identities, which is not comparable. （2）Darblur uses a blurring method with the aim of generating blurred results, which is still not comparable to the method in this paper.  （3）Some of the advanced methods of face privacy protection [6,7]  can go for comparison.

     The experimental section describes flexible (personalized) protection using only a small amount of content, which is unreasonable. The de-identification performance in comparison with other methods should be scaled down and the validation of flexible protection should be increased.

     [6] 2021TCSVT-The UU-Net_ Reversible Face De-Identification for Visual Surveillance Video Footage

     [7]2021ICCVPersonalized and invertible face de-identification by disentangled identity information manipulation
5.  **Protected faces in the real world can be displayed.** Overall, the work in this paper is more of a new method for face manipulation. It is also crucial for face privacy protection that the method can be effective in the real world. Therefore, adding real-world face image protected results can enhance the persuasiveness of this paper.

**Suitability:**

3

---

### Official Review · Reviewer_t6bM · 2024-05-07

**Rating:** 4
**Confidence:** 3

**Summary:**

This paper proposes a two-stage latent representation reorganization (LReOrg) framework to protect face privacy. The sensitive information is anonymized in the first stage, and attribute information is recovered according to usability requirements. LReOrg provides flexible control by manipulating which attributes to anonymize or preserve. Specifically, this paper constructs a novel Conditional Disentanglement Fusion Network (CDFNet) to separate sensitive information for protection based on keywords. Different anonymization strategies are employed for identity and attributes: Group-based Identity Anonymizer (GIA) and k-farthest Attribute Anonymizer (KAA).

**Strengths:**

Fine-grained attributes control is crucial in the field of privacy protection. This paper is innovative, both in its ideas and its technologies. In many details, this paper exhibits unique insights. Adequate experimental have validated the effectiveness of the method.

**Limitations:**

1. The Group-based Identity Anonymizer (GIA) employs a differential privacy strategy based on the exponential mechanism. However, this strategy does not satisfy the post-processing property of differential privacy during the recovery reorganization stage. The premise of post-processing is that it does not involve original data. However, in the recovery stage, the use of $s_x^0$ extracted from the original data results in excessive consumption of privacy budget.
2. CLIP provide an interpretation of the image, but their interpretability for facial attributes is limited due to the lack of specialized training. Using standard CLIP as a discriminator in Eq.10 is not a good choice.
3. Is the training process of CDFNet randomly selecting $k_i$ (identity and attributes) for training in each epoch? If so, would this lead to the distance of CLIP being influenced by identity? Specifically, at a certain point, when the network already possesses identity-preserving capabilities, $I_2'$ preserves the identity of $I_1$, and $I_1'$ preserves the identity of $I_2$. CLIP erroneously believes it has preserved the attribute ('gender') in attribute-oriented swap (AOS) training due to the correlation between identity and attributes.
4. It is suggested to clarify that $D(·)$ in Eq. 10 denotes the CLIP discriminator, representing the similarity between images and text.
5. Is Figure 7 a step-by-step process, where the fourth column changes gender based on the third column, and the fifth column changes expression based on the fourth column? If so, it is advisable to add explanations in the paper.
6. The identity recovery ability suffers from significant drops. The reason might be that there are too many identity categories while attributes are only a few.

**Suitability:**

3

---

### Official Review · Reviewer_WLXr · 2024-05-24

**Rating:** 4
**Confidence:** 4

**Summary:**

This paper presents a generative framework that protects the identity and sensitive attributes of face images. The proposed method employs a two-stage recurrent grafting to de-identify and recover chosen attributes. Experimental results show the method offers a desirable level of privacy protection while retaining task utility.

**Strengths:**

- Interesting research topic: Face protection is an important topic and is worth further exploration.
- The proposed method is well-grounded and achieves its protection goal.
- The experiments are extensive.

**Limitations:**

It is a coincidence that I reviewed the same paper at ACM MM last year (it was titled "ReGraft: A Recurrent Grafting Framework for Personalized Face Privacy Protection"). I liked this paper overall albeit some minor performance issues and my last year's final rating was borderline acceptance. It regretfully wasn't finally accepted. I am delighted to see this paper is further polished, and the authors have incorporated some of my previous suggestions (e.g., comparison with diffusion-based methods, further analysis on facial attribute protection, and additional coverage of related works). I am not going to raise further concerns as they are mostly already addressed through the authors' revision. I keep my last year's rating.

**Suitability:**

3

---

### Official Review · Reviewer_iQGQ · 2024-05-24

**Rating:** 3
**Confidence:** 1

**Summary:**

The paper proposes a face anonymizing technique based on latent vector manipulation of the style transfer model. The method enables inputting the face attributes and dividing them as sensitive and non-sensitive. the sensitive attributes anonymization is carried out. Many experiments are carried out to show the face quality generation and the possibility

**Strengths:**

- many experiments were done to evaluate face generation and privacy protection. This effort is appreciated.
- The idea of dividing face attributes into sensitive and non-sensitive provides flexibility in face anonymization.

**Limitations:**

- The method proposed is performing face attribute editing. This is a well-explored area with numerous existing works [1, 2], and the technical novelty of the proposed method appears limited.
- I don't understand the need for using CLIP. Why is it necessary to use CLIP to encode keywords instead of directly using attribute indices as values and embedding them into vectors?
- The criteria for distinguishing between sensitive and non-sensitive attributes are not clearly defined. Providing a detailed explanation of this selection process would enhance the paper's clarity.
- The application of group differential privacy is not sufficiently explained. A more thorough description of its implementation is necessary to understand its role and effectiveness in the proposed method.


- Other comments:
1) The method presentation is poor. Fig. 2, 3 and 4 are hard to understand, and the captions do not provide necessary information.
2) The writing of the paper is not the best. It can be improved.


[1] Qiu, Haonan, et al. "Semanticadv: Generating adversarial examples via attribute-conditioned image editing." Computer Vision–ECCV 2020: 16th European Conference, Glasgow, UK, August 23–28, 2020.
[2] Ding, Zheng, et al. "Diffusionrig: Learning personalized priors for facial appearance editing." Proceedings of the IEEE/CVF Conference on Computer Vision and Pattern Recognition. 2023.

**Suitability:**

3

---

### Meta-Review · Area_Chair_iKR7 · 2024-06-28

**Recommendation:** Accept (Poster)
**Confidence:** 5

**Metareview:**

This paper presents a generative framework that protects the identity and sensitive attributes of face images. The proposed method employs a two-stage recurrent grafting to de-identify and recover chosen attributes. Experimental results show the method offers a desirable level of privacy protection while retaining task utility.

Initially, this paper receives mixed reviews with 2 BR and 2 BA. The authors have well addressed the reviewer concerns. After the rebuttal, all the reviewers give BA score. After carefully checking the  reviewer comments, this work has an interesting method of solving problems, and the experiment is adequate.

I would like to accept this paper. The authors are encouraged to carefully prepare the camera ready version according to the reviewer recommendations.

---

### Meta-Review · Senior_Area_Chairs · 2024-07-10

**Recommendation:** Accept (Poster)
**Confidence:** 4

**Metareview:**

This paper received mixed ratings initially. After rebuttal, all the reviewers tend to accept the paper. SAC and AC agree with reviewers and recommend acceptance of the paper.